# Oleracone F Alleviates Cognitive Impairment and Neuropathology in APPswe/PSEN1dE9 Mice by Reducing the Expression of Vascular Cell Adhesion Molecule and Leukocyte Adhesion to Brain Vascular Endothelial Cells

**DOI:** 10.3390/ijms24032056

**Published:** 2023-01-20

**Authors:** Young-Sun Kwon, Jin-Sung Ko, Se-Young Oh, Young Taek Han, Sangmee Ahn Jo

**Affiliations:** 1Department of Nanobiomedical Science & BK21 Four NBM Global Research Center for Regenerative Medicine, Dankook University, Cheonan 31116, Republic of Korea; 2Department of Convergence Medicine, Ewha Womans University Mokdong Hospital, Ewha Womans University, Seoul 07985, Republic of Korea; 3Department of Pharmacy, College of Pharmacy, Dankook University, Cheonan 31116, Republic of Korea

**Keywords:** oleracone, vascular cell adhesion molecule-1, tumor necrosis factor-α, Alzheimer’s disease, blood–brain barrier

## Abstract

Alzheimer’s disease (AD) is the most common neurodegenerative disease and the blood–brain barrier dysfunction has been suggested as a key pathological feature of the disease. Our research group successfully established a synthetic protocol for oleracones, a novel series of flavonoids isolated from the plant extract of *Portulaca oleracea* L. (PO). PO extract was reported to have anti-inflammatory and antioxidant effects, enhancing cognitive function. Thus, we investigated the effects and mechanism of oleracones on cognition using AD model transgenic mice (Tg; APPswe/PSEN1dE9). Oleracone F treatment significantly improved memory dysfunction in Tg mice. Oleracone F decreased the number, burden, and immunoreactivity of amyloid plaques and amyloid precursor protein (APP) protein levels in the brains of Tg mice compared to wild-type mice. Oleracone F also alleviated inflammation observed in Tg mice brains. In vitro studies in human microvascular endothelial cells (HBMVECs) demonstrated that oleracones D, E, and F blocked the elevations in VCAM-1 protein induced by tumor necrosis factor-α (TNF-α), hindering leukocyte adhesion to HBMVECs. Taken together, our results suggest that oleracones ameliorated cognitive impairment by blocking TNF-α-induced increases in VCAM-1, thereby reducing leukocyte infiltration to the brain and modulating brain inflammation.

## 1. Introduction

Alzheimer’s disease (AD) is a typical progressive neurodegenerative disorder characterized by neuronal loss leading to brain dysfunction, including memory impairment [1]. The neuropathological features of AD are beta-amyloid (Aβ) accumulation, tau hyperphosphorylation, brain inflammation, gliosis, and the loss of synapses [2,3,4,5]. Recent evidence demonstrated that dysfunction of the blood–brain barrier (BBB) was an early feature in prodromal AD. BBB breakdown at an early age has been observed in the hippocampal regions of individuals with mild cognitive impairment [6], and in several gray and white matter regions in patients in the early stages of AD [7]. Another factor of BBB dysfunction that is closely involved in the initiation and/or progression of neurodegenerative events in AD is the induction of adhesion proteins by systemic and vascular inflammation [8], all of which facilitate the transmigration of circulating leukocytes.

In response to various stimuli such as infection, hypertension, and high glucose blood levels, brain endothelial cells produce free oxygen radicals and inflammatory cytokines, such as tumor necrosis factor-α (TNF-α). These inflammatory mediators activate endothelial cells to upregulate the expression of cell adhesion molecules (CAMs), including intercellular adhesion molecule-1 (ICAM-1), vascular cell adhesion molecule-1 (VCAM-1), and E-selectins [9]. These adhesion proteins provide attachment sites for blood-circulating leukocytes to transmigrate through vascular endothelial walls to reach the target tissue. During the transmigration process, the leukocytes produce and release reactive oxygen species, proteases, and other inflammatory substances that could potentially damage microvessels [10], which would further induce the recruitment of other inflammatory mediators at the sites. This would eventually cause uncontrollable brain inflammation [11], leading to the pathogenesis of many brain diseases, such as bacterial meningitis [12], stroke [13], multiple sclerosis [14], and AD [15]. Thus, blocking leukocyte transmigration into the brain could be an important strategy for preventing and treating numerous central nervous system diseases [16].

Many folk remedies have been reported to alleviate AD progression. *Portulaca oleracea* L. (PO), a well-known weed widely distributed in tropical and temperate regions, has been used for food and as a folk remedy to relieve a variety of diseases due to its wide spectrum of pharmacological properties, including antioxidant, anti-inflammatory, anti-microbial, anti-diabetes, anti-ulcerogenic, anticancer, and neuroprotective activities [17]. However, only a few molecular mechanisms of action of this compound are known. An aqueous extract of PO (AP) showed anti-TNF-α activity in vascular endothelial cells [18], suggesting possible anti-vascular inflammatory effects. AP has been reported to attenuate the TNF-α-induced upregulation of ICAM-1, VCAM-1, and E-selectin in human umbilical vein endothelial cells (HUVECs). However, the active component(s) responsible for these responses is not yet known. PO contains diverse compounds, such as flavonoids, alkaloids, coumarins, polysaccharides, fatty acids, sterols, and monoterpene glycosides [18]. Oleracones are a new family of flavonoids isolated from PO [17]. Eight types of oleracone (A, B [19], C, D, E [17], F [20], J, and K [21]) have been isolated and recently, oleracones D-F were successfully synthesized [22]. Little is known about the pharmacological activities of oleracones, but oleracone C, D, E, F, and G presented scavenging activities in the 1,1-diphenyl-2-picryl-hydrazyl (DPPH) radical-quenching assay, as well as anti-inflammatory antioxidant, and anti-cholinesterase activities [17,20,21]. Considering their pharmacological effects in alleviating inflammatory conditions, this study aimed to investigate the effects of oleracones on vascular dysfunction and memory loss using AD model transgenic (Tg) mice.

## 2. Results

### 2.1. Effect of Oleracones on Cell Viability

Oleracones D, E, and F were synthesized as described in the Methods. The structures of the oleracones are shown in Figure 1A. To examine the cellular toxicity of oleracones, human microvascular endothelial cells (HBMVECs) were treated with various concentrations (0–100 μM) of oleracones for 24 and 72 h, and cell viability was determined using the MTT assay. After 24 h, cell viability was not significantly influenced by the three oleracones at concentrations of up to 25 μM (Figure 1B) but viability decreased at concentrations higher than 50 μM. Similar results were observed when the cells were treated with oleracones for 72 h (Appendix A). These results indicate that none of the three oleracones were toxic to HBMVECs at concentrations lower than 25 μM. Oleracone E was more toxic than oleracones D and F at concentrations higher than 50 μM. Thus, sublethal concentrations up to 25 μM were used in the subsequent experiments.

### 2.2. Oleracones Blocked TNF-α-Induced Increases in Leukocyte Adhesion to Endothelial Cells and VCAM-1 Protein Expression 

To investigate the role of oleracones in mitigating endothelial dysfunction, we examined the effect of oleracones on HL-60 leukocyte adhesion to HBMVEC monolayers treated with TNF-α. As shown in Figure 2, treatment with TNF-α (10 ng/ML) significantly increased the adhesion of HL-60 leukocytes to the endothelial monolayer. However, pretreatment with oleracones D, E, and F significantly suppressed TNF-α-induced leukocyte adhesion to the monolayer in a concentration-dependent manner. The half-maximal inhibitory concentrations (IC_50_) of oleracones D, E, and F were approximately 6.1, 5.2, and 6.4 μM, respectively.

The effect of oleracones on the EC adhesion molecules ICAM-1 and VCAM-1 induced by TNF-α was evaluated by Western blotting analysis because these two adhesion proteins play an important role in the adhesion of leukocytes to endothelial cells. As shown in Figure 2B,C, the treatment of HBMVECs with TNF-α (10 ng/mL) dramatically increased the protein levels of both ICAM-1 and VCAM-1. However, oleracones D, E, and F significantly blocked TNF-α-induced VCAM-1 protein in a concentration-dependent manner without having a significant effect on ICAM-1 protein (Figure 2B,C).

### 2.3. Oleracone F Attenuated Cognitive Impairment in AD Model Mice

To examine the effect of oleracones on cognitive function, oleracone F was selected among three oleracones because of its low toxicity. Oleracone F or vehicle (a mixture of saline and 10% Cremophor EL) was administered intraperitoneally (i.p) (5 mg/kg, in 200 μL) to 10-month-old Tg and wild-type (WT) mice. The experiments were conducted according to the scheme shown in Figure 3A. During 4 weeks of oleracone F treatment, body weight was evaluated and no significant difference between the oleracone F and vehicle groups was found throughout the experimental period. Similar results were obtained for brain weight between the two groups. These results indicate that oleracone F administration at the dose applied in this study was not likely to cause any critical toxicity in the mice.

Then, the effect of oleracone F on cognitive function was examined by the passive avoidance test. Initially, the latency time (training latency) of the mice before the electric shock was examined. A mild electric shock was applied to the mice, and the latency time was measured 24 h later. As shown in Figure 3D, the initial training latency was not significantly different between the groups. In contrast, the latency time measured after shocking was significantly reduced in vehicle-Tg compared to vehicle-WT mice (vehicle-Tg 60.7 ± 28.0 vs. vehicle-WT 411.2 ± 170.1, *p =* 0.0003), but this reduction was prevented by oleracone F treatment (vehicle-Tg 60.7 ± 28.0 vs. oleracone-Tg 282.0 ± 143.2, *p =* 0.0092). There was no significant difference between vehicle- and oleracone F-treated WT mice.

### 2.4. Oleracone F Blocked VCAM-1 Expression in the Vessels of Tg Mice

To evaluate the role of VCAM-1 in AD pathogenesis and the effect of oleracones on VCAM-1 expression, the expression of VCAM-1 in the brains of Tg mice administrated with oleracone F was analyzed. As expected, the Tg mice brains had higher VCAM-1 densely localized along the lining of the brain vessels (red) compared to WT mice brains (Figure 4A; vehicle-WT 33.3 ± 7.5 vs. vehicle-Tg 88.3 ± 0.70, *p =* 0.018), suggesting the induction of VCAM-1 in the Tg brain vessels. Moreover, oleracone F administration prevented VCAM-1 induction in Tg brains (vehicle-Tg 88.3 ± 0.70 vs. oleracone F-Tg 49.4 ± 7.0, *p =* 0.031). Similar results were obtained, where Tg mice had higher protein levels of cortical and hippocampal VCAM-1 compared to WT mice (cortex, vehicle-Tg 1.7 ± 0.1 vs. vehicle-WT 1.0 ± 0.010, *p* = 0.0009; hippocampus, vehicle-Tg 1.5 ± 0.031 vs. vehicle-WT 1.0 ± 0.010, *p =* 0.0001), and this was prevented by oleracone F treatment (cortex, vehicle-Tg 1.7 ± 0.12 vs. oleracone F-Tg 1.1 n 0.051, *p* = 0.034; hippocampus, vehicle-Tg 1.5 ± 0.031 vs. oleracone F-Tg 1.1 ± 0.044, *p =* 0.002).

### 2.5. The Effect of Oleracone F on the Levels of Brain Amyloids and APP Protein in Tg Mice

Amyloid plaques are the hallmark of the neuropathological characteristic of AD. Thus, the effect of oleracone F on brain Aβ levels was examined by immunofluorescence staining. As shown in Figure 5A, Aβ was detected in both the cortex and hippocampus of Tg mice but not in the brains of WT mice. Oleracone F administration to Tg mice significantly prevented Aβ plaque formation compared with those in vehicle-treated Tg mice. There were significant reductions in the number of Aβ plaques (cortex, vehicle-Tg 77.0 ± 3.0 vs. oleracone F-Tg 38.0 ± 8.0, *p* = 0.045; hippocampus, vehicle-Tg 57.0 ± 7.0 vs. oleracone F-Tg 19.5 ± 4.5, *p* = 0.046), Aβ plaque burden (cortex, vehicle-Tg 5.1 ± 0.070 vs. oleracone F-Tg 2.4 ± 0.70, *p* = 0.002), and Aβ plaque fluorescence intensity (cortex, vehicle-Tg 100.5 ± 9.1 vs. oleracone F-Tg 70.5 ± 0.16, *p* = 0.004; hippocampus, vehicle-Tg 100.0 ± 0.080 vs. oleracone F-Tg 63.3 ± 9.70, *p* = 0.002) in the cortex and hippocampal regions of the brains from Tg mice treated with oleracone F compared to vehicle-treated mice.

Since Aβ is synthesized from amyloid precursor protein (APP) the effect of oleracones on APP protein levels was also analyzed. Human APP protein was detected in both the cortex and hippocampus of Tg mice but not in those of WT mice (Figure 5B). The human APP protein levels in oleracone F-treated Tg mice were significantly reduced compared to vehicle-treated Tg mice (cortex, vehicle-Tg 16.7 ± 1.86 vs. oleracone F-Tg 8.1 ± 1.71, *p* = 0.027; hippocampus, vehicle-Tg 32.5 ± 3.31 vs. oleracone F-Tg 18.1 ± 2.67, *p* = 0.042).

### 2.6. Effect of Oleracone F on Brain Inflammation in Tg Mice

The effect of oleracone F on brain inflammation was examined by measuring the protein levels of glial fibrillary acidic protein (GFAP) and ionized calcium-binding adapter molecule 1 (Iba-1), which are known as markers of activated astrocytes and microglia cells, respectively. The expression of both GFAP and Iba-1 in the cortex of Tg mice was increased compared to that of WT, but this induction was significantly prevented by oleracone F treatment (Figure 6A) (GFAP, vehicle-Tg 3.8 ± 0.39 vs. oleracone F-Tg 2.2 ± 0.12, *p* = 0.049; Iba1, vehicle-Tg 1.8 ± 0.081 vs. oleracone F-Tg 1.1 ± 0.082, *p* = 0.015). Similar results were found in the hippocampus (GFAP, vehicle-Tg 2.2 ± 0.30 vs. oleracone F-Tg 2.1 ± 0.19, *p =* 0.015; Iba1, vehicle-Tg 1.1 ± 0.044 vs. oleracone F-Tg 0.79 ± 0.085, *p* = 0.018).

## 3. Discussion

PO has been used as a folk medicine that has various beneficial roles in treating inflammation and neurological disorders. Recent evidence suggests that aqueous PO extract also protected against lipopolysaccharide-induced memory loss in mice [23]. PO extract prevented the vascular inflammation process by inhibiting intracellular ROS production, as well as reducing protein adhesion induced by TNF-α treatment in HUVECs [18]. Thus, it is possible to speculate that PO extract might delay neurodegeneration progression by regulating vascular inflammation responses, such as the induction of adhesion proteins and the generation of ROS. In previous studies, PO extract prevented the induction of VCAM-1, ICAM-1, and E-selectin proteins mediated by TNF-α, but the principal bioactive components of the PO extracts responsible for this activity were not identified [18]. In the present study, we identified oleracones D, E, and F, the major flavonoid compounds of the PO extract that specifically target VCAM-1, downregulating the expression of endothelial VCAM-1 but not ICAM-1, E-selectin, or PECAM-1 (Figure 2 and Appendix A). Thus, these findings suggest oleracones D-F as specific modulators of VCAM-1. Since it is generally known that the NF-κB signaling pathway co-regulates the gene expression of both ICAM-1 and VCAM-1 [22,24], the effect of oleracones solely on VCAM-1 suggests the involvement of independent regulatory pathways that differentially regulate ICAM-1 and VCAM-1 apart from the NF-κB signaling pathway.

Among the oleracone isotypes that specifically target endothelial VCAM-1, oleracone F was selected among three oleracones because of its low toxicity. Therefore, we selected oleracone F to determine its potential application as a compound that alleviates AD conditions using AD Tg mice. As speculated, the Tg mice had higher levels of VCAM-1 in the brain vessels located in the cortex, hippocampus (Figure 4), and other brain areas, and oleracone F treatment significantly reduced VCAM-1 levels. We previously demonstrated the presence of leukocytes attached to brain vessels and infiltrated into brain parenchyma in Tg mice brains [25]. A similar report in the brains of AD patients was made, where circulating leukocytes (neutrophils and T cells) adhered to the activated endothelium of cerebral vessels, damaging the parenchyma of the brain [6,7,26,27]. Thus, the results found in our current study suggest a close association between VCAM-1 induction and AD development, probably through the enhanced infiltration of leukocytes. The role of leukocyte infiltration as a trigger of brain inflammation and AD development has been proposed in other studies [15]. Therefore, the regulation of vascular inflammation could be considered a therapeutic target for delaying AD progression. For example, there are reports that blocking neutrophil adhesion to endothelial cells with an antibody against lymphocyte function-associated antigen 1, the receptor for ICAMs, ameliorated cognitive loss in Tg mice [15] and that anti-VCAM-1 antibodies attenuated atherosclerosis in Apolipoprotein E^−/−^ mice by inhibiting the adhesion of inflammatory cells [28]. In the present study, we also demonstrated the enhanced inflammation in the brains of AD mice, which could be diminished by oleracone administration. 

From our results, we propose the mechanism underlying the effect of oleracones on cognitive function. In AD Tg mice, elevated amounts of proinflammatory cytokines, such as TNF-α and IL-1β, are released by microvessels of the brain [29]. Then, the released inflammatory cytokines activate endothelial cells to increase the protein expression of adhesion molecules such as VCAM-1, enhancing leukocyte recruitment. The infiltrated immune cells in the brain dramatically induce the activation of brain proinflammatory response in the brain, as evidenced by the increased expression of GFAP and IBA. In addition, increases in VCAM-1 could activate the amyloid synthesis pathway, and/or suppress the amyloid degradation pathways as evidenced by both the increased APP level and increased Aβ plaque numbers, Aβ plaque burden. Interestingly, oleracones decreased the progression of all these AD pathologies by inhibiting the expression of VCAM-1. Thus, it seems reasonable to speculate that the inhibition of adhesion proteins could be an ideal therapeutic target for treating AD.

Studies have shown that flavonoids activated antioxidant pathways that render an anti-inflammatory effect. They inhibit the secretions of enzymes, such as lysozymes and arachidonic acid, which reduce inflammatory reactions and modulate the expression and activation of proinflammatory cytokines, such as TNF-α, interleukin-1β, interleukin-6, and interleukin-8 [30]. Recent publications showed that several flavonoids could reduce the expression of adhesion proteins through another mode of action. Formononetin, an isoflavone, has been shown to attenuate the Aβ_25–35_-induced expression of ICAM-1 and VCAM-1 in HBMVECs. Chrysin, a flavonoid component of propolis, decreased LPS-induced VCAM-1 expression [31] in mouse cerebral vascular endothelial cells (bEnd.3). Moreover, memantine, an N-methyl-D-aspartate receptor antagonist that is therapeutically prescribed for the moderate-to-severe stages of AD [32], was shown to reduce the expression of CAMs, exhibiting a protective role against vascular pathology. The results of our study also showed that oleracone F reduced vascular pathology by downregulating VCAM-1 in endothelial cells, thereby inhibiting leukocyte adhesion to vessels induced by inflammation. Thus, it could potentially be used as a treatment to slow AD progression.

## 4. Materials and Methods

### 4.1. Materials

Culture media M199, fetal bovine serum (FBS), and antibiotics were purchased from Gibco (Carlsbad, CA, USA). Normal goat serum, mounting medium, and Dylight^®^ 594 anti-rabbit IgG were purchased from Vector Laboratories (Burlingame, CA, USA). Alexa Fluor 488 goat anti-mouse IgG antibody was acquired from Molecular Probes (Eugene, OR, USA). Rabbit polyclonal antibodies were purchased from Santa Cruz Biotechnology (Santa Cruz, CA, USA). Human TNF-α, ethylenediaminetetraacetic acid, ethylene glycoltetraacetic acid, phenylmethylsulfonyl fluoride (PMSF), and Triton X-100 were purchased from Sigma (St. Louis, MO, USA). The primary antibodies used in this study are summarized in Table 1. Oleracones D, E, and F were synthesized in Professor Han’s laboratory as previously described [33].

### 4.2. Cell Culture

Human brain microvascular endothelial cells (HBMVECs) were purchased from Cell Systems (Kirkland, WA, USA) and grown on attachment factor-coated plates in CSC complete serum-free medium (Cell Systems, Seattle, WA, USA) or in M199 medium supplemented with 20% FBS, 5 U/mL of heparin, 3 ng/mL of recombinant human fibroblast growth factor-basic (FGF-β; Millipore, Temecula, CA, USA), penicillin (100 U/mL), and streptomycin (100 μg /mL) at 37 °C in an atmosphere of 5% CO_2_ in air. Human leukemia-60 (HL-60) cells were purchased from the Korean Cell Line Bank (Seoul, Korea) and maintained in RPMI 1640 supplemented with 10% FBS, penicillin (100 U/mL), and streptomycin (100 μg/mL).

### 4.3. Cell Adhesion Assay

An adhesion assay was performed as previously described [34]. In brief, confluent HBMVEC monolayers in 12-well plates were pretreated with oleracones at 37 °C for 1 h followed by treatment with TNF-α (10 ng/mL) for 8 h. After the treatments, the cells were washed with warm Dulbecco’s phosphate buffered saline (DPBS), and HL-60 cells (1 × 10^5^) in 0.5 mL RPMI 1640 were added to the HBMVEC monolayers and incubated in a CO_2_ incubator at 37 °C for 1 h to allow the attachment of the HL-60 cells to the monolayers. The HBMVECs were washed three times with DPBS to remove non-adherent and loosely attached HL-60 cells. The attached HL-60 cells were counted under an inverted microscope at 200× magnification.

### 4.4. Cell Toxicity Assay

3-(4,5-Dimethylthiazole-2-yl)-2,5-diphenyl tetrazolium bromide (MTT, Sigma, St Louis, CA, USA) was used to evaluate the effect of oleracones on cell toxicity. Briefly, HBMVECs were cultured in 96-well plates at a density of 1.0 × 10^4^ cells/well. Then, MTT dissolved in media was added to each well at a final concentration of 0.5 mg/mL and incubated at 37 °C for 24 h or 72 h. The culture media was removed carefully and 100 μL of dimethyl sulfoxide (DMSO, Sigma, St. Louis, CA, USA) was added to dissolve the formazan crystals. The absorbance of the MTT product was measured using a Spectro Plate Reader (Bio Tek, Winooski, VT, USA) at 570 nm according to the manufacturer’s instructions.

### 4.5. Transgenic Mice and Genotyping

All experimental procedures and methods involving animals were approved by Dankook University Animal Care and Use Committee (1D: 13-033). All mice used in this study were housed in individual cages under temperature- and light-controlled (12 h light/12 h dark cycle) specific pathogen-free conditions. They were allowed free access to standard irradiated chow (Purina Mills, Seoul, Republic of Korea) and water.

AD model Tg mice (APPswe/PSEN1dE9) were purchased from Jackson Laboratory (Bar Harbor, ME, USA). These mice express chimeric mouse/human amyloid precursor protein (Mo/HuAPP695swedish) and mutant human presenilin 1 (PSEN1 deletion exon 9). The humanized Mo/HuAPP695swe transgene allows the mice to secrete a human A-beta peptide. They were maintained as double-hemizygotes by crossing with C57BL6 wild-type mice. For genotyping, genomic DNA was extracted from mice tails using an EzDirect^TM^ Mouse Direct PCR kit (Wizbiosolutions Inc., Gyeonggi-do, Republic of Korea) according to the protocol provided by the company. Positive-Tg genotype was detected by polymerase chain reaction (PCR) analysis using the following primers: *human PS1*-forward, 5′-AAT AGA GAA CGG CAG GAG CA-3′; and *human PS1*-reverse, 5′-GCC ATG AGG GCA CTA ATC AT-3′.

### 4.6. Drug Administration

Oleracone F was dissolved in DMSO and then in a vehicle solution, a mixture of saline and 10% Cremophor EL (EMD Millipore Corp., Darmstadt, Germany). The WT and Tg mice were divided into two groups: the vehicle and the oleracone F-administrated groups. The mice were administered vehicle or oleracone F at a dose of 5 mg/kg by intraperitoneal injection (i.p), daily for 28 days.

### 4.7. Passive Avoidance Test

Passive avoidance is a fear-motivated test to study long-term memory in an associative manner using a shuttle box device that consists of two compartments separated by a retractable door. One compartment has dark opaque walls and a roof, and another compartment has bright lighting. The floor in both compartments is made of shocking metal grids, except that the floor is wired to receive a mild electric shock of 0.3 mA intensity for a 1-s duration on the unsafe side.

The experiment was conducted for two days. On the first day, the mouse was allowed to adapt while freely moving around the box separated into two compartments for 5 min. Then, the mouse was removed from the box and returned to the cage. The box was cleaned with 70% ethanol for each use. After adaptation, the mouse was placed in a bright area for 30 s, and the door to the dark room was opened for the mouse to enter. The initial latency time for the mouse to move into the dark room was measured for up to 5 min. When the mouse entered the dark room, the door was closed, and after 5 s, an electric shock of 0.3 mA was applied through the floor grid for 1 s, followed by a 1 min pause. The mouse was removed from the box. On the second day, the mouse was placed in a bright area, and the door to the dark room was opened 30 s later. The time of entering the dark room was measured for up to 10 min.

### 4.8. Brain Section Preparation

Mice were anesthetized and perfused with 10 mM PBS, pH 7.4, and their brains were isolated and cut in half. For the immunofluorescence assay, the right hemisphere was fixed with 4% paraformaldehyde prepared in PBS at 4 °C overnight and then preserved in 30% sucrose prepared in PBS until reaching equilibrium. Then, the brains were frozen in OCT compound and sectioned in 20-mm thick slices using a Cryo-cut machine (Leica, Wetzlar, Germany). The sections were stored in a storage buffer (glycerol 30%, ethylene glycerol 30% in 10 mM PBS) and kept at −80 °C. For Western blotting analysis, the cortex and hippocampus were isolated from the left hemisphere, rapidly frozen in liquid nitrogen, placed in a 1.5 mL microtube, and stored in a −70 °C freezer.

### 4.9. Immunofluorescence Staining

Sections were rinsed with PBS, permeabilized with 0.05% Triton-100, and blocked with 5% normal goat serum (Vector Laboratory, Burlingame, CA, USA) prepared in 0.01 M PBS buffer containing 0.05% TritonX-100 for 1 h at room temperature. The samples were incubated with primary antibodies at 4 °C overnight. The primary antibodies used in this study are listed in Table 1. After three washes with 0.01 M PBS, the sections were incubated with specific secondary anti-mouse Alexa Fluor 488 (Life Technologies, Carlsbad, CA, USA) or anti-rabbit Dylight 594 (Vector Laboratories, Inc., CA, USA) antibodies at room temperature for 1 h in the dark. After washing with PBS three times, the sections were mounted using a mounting medium containing 4′, 6-diamidino-2-phenylindole (DAPI; Vector Laboratories, Inc., CA, USA). Immunoreactivity in the brain was assessed using a ZEN 2009 software on a Zeiss LSM confocal microscope (Carl Zeiss, Jena, Germany) equipped with a 400× objective lens.

### 4.10. Western Blot Analysis

For the protein analysis of brain tissues, the cortex and hippocampus were isolated from mice and lysed with RIPA extraction buffer containing 20 mM Tris-HCl pH 7.4, 150 mM NaCl, 1 mM ethylenediaminetetraacetic acid, 1% Triton X-100, 1 mM ethylene glycol tetraacetic acid, protease inhibitor cocktail tablets (PIC, Roche, Basel, Switzerland), protein phosphatase inhibitor cocktail (PPIC, Roche, Basel, Switzerland), and 1 mM PMSF by sonication (for 30 s, pulse: 10, 5 s, amplitude: 30%) while keeping the tube in a beaker containing ice. Then, the extract was further incubated for 40 min in ice. Cultured HBMVECs were washed with PBS and lysed with lysis buffer [20 mM Tris-HCl (pH 7.5), 150 mM NaCl, 1% Triton X-100, 1 mM EDTA, 1 mM EGTA] containing 1 mM PIC, 1 mM PPIC, and 1 mM PMSF. The supernatants were separated by centrifugation (21,000× *g*, 30 min, 4 °C). Protein concentrations were determined using a BCA protein assay kit (Pierce, Rockford, IL, USA). Equal quantities of proteins (20–60 µg) were dissolved in sample buffer [50 mM Tris-HCl pH 6.8, 2% sodium dodecyl sulfate (SDS), 100 mM dithiothreitol, 0.01% bromophenol blue, and 10% glycerol], separated by SDS-polyacrylamide gel electrophoresis, and transferred to polyvinylidene difluoride membranes. The blots were blocked using 5% skim milk and then incubated with the appropriate primary antibodies listed in Table 1 at 4 °C overnight and appropriate horseradish peroxidase-conjugated IgG anti-mouse or anti-rabbit secondary antibodies (Santa Cruz Biotechnology, Dallas, TX, USA) at room temperature for 1 h. The blots were finally developed using enhanced chemiluminescence reagents (ECL, Advanta; Menlo Park, CA, USA). The expression level of each protein band was quantified using ImageJ Software (National Institutes of Health, Bethesda, MD, USA).

### 4.11. Quantification and Statistical Analysis

All results are expressed as the mean ± standard error of the mean (SEM). Data were analyzed by one-way ANOVA followed by post-hoc Dunnett’s multiple comparisons test and significant differences are denoted as * *p* < 0.05, ** *p* < 0.01, or *** *p* < 0.001. A Student’s *t*-test was applied for the analysis of significant differences between the two groups. Calculations were performed using GraphPad Prism 7 software (GraphPad Software, Inc., San Diego, CA, USA).

## 5. Conclusions

Vascular inflammation induced by proinflammatory cytokines has been shown to induce endothelial dysfunction, leading to the development of various neurodegenerative diseases, including AD. Our study demonstrated that oleracone F, a novel flavonoid isolated from PO, alleviated decreases in cognitive function in AD Tg mice. Oleracone F treatment also decreased APP protein expression, Aβ plaque accumulation, and inflammatory pathologies, such as GFAP and Iba1 in the brains of Tg mice. Additional results revealed that these beneficial effects of oleracone F were likely due to reductions in VCAM-1 expression in endothelial cells, which decreased leukocyte adhesion to HBMVECs. Taken together, our results suggest that oleracones ameliorated cognitive impairment by blocking TNF-α-induced increases in VCAM-1, thereby reducing leukocyte infiltration to the brain and modulating brain inflammation. Our study provides important evidence of the beneficial effects of oleracones for AD therapy and suggests oleracone F as a promising candidate for AD therapeutics.

## Figures and Tables

**Figure 1 ijms-24-02056-f001:**
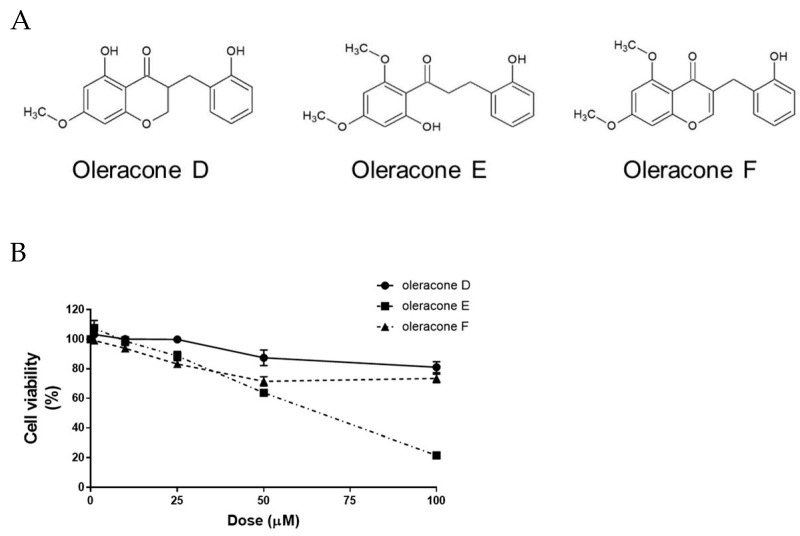
Structure and the effects of oleracones on cell viability. (**A**) The structure of oleracones D, E, and F are shown. (**B**) HBMVECs were pretreated with oleracones at the concentrations indicated 1 h prior to treatment with TNF-α (10 ng/mL, for 24 h). The cell viability was determined by the MTT assay.

**Figure 2 ijms-24-02056-f002:**
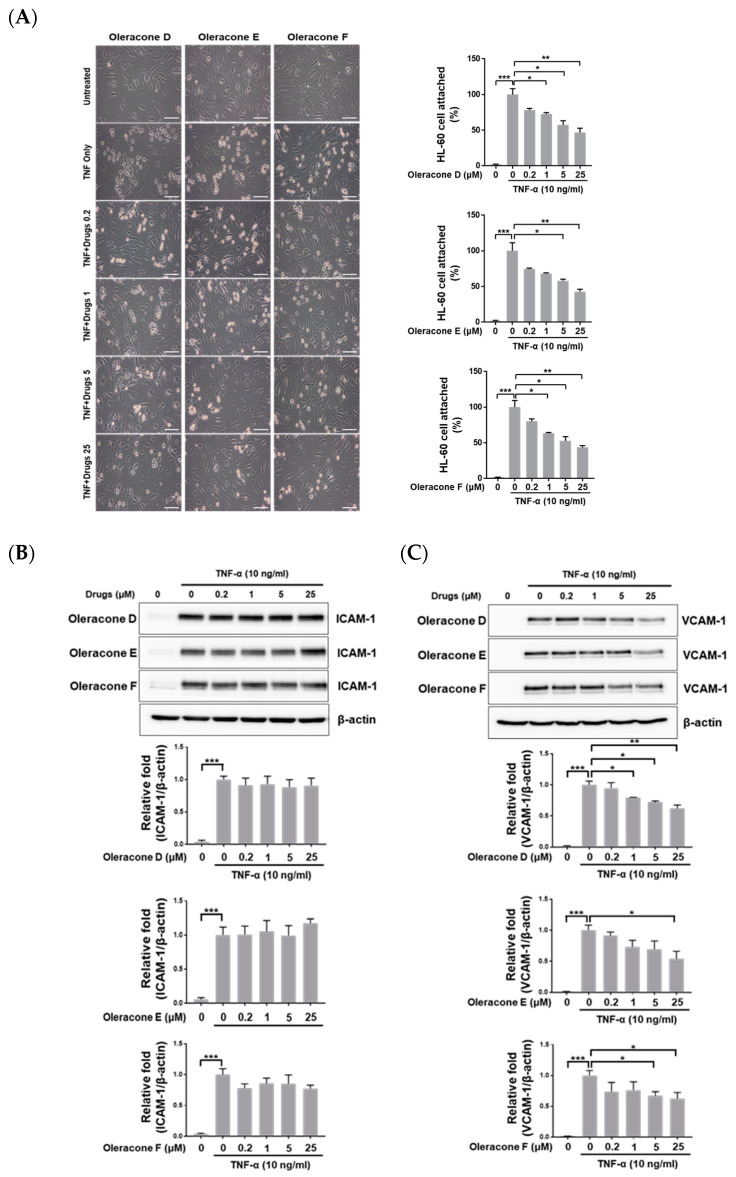
Oleracone F blocked the leukocyte adhesion and the expression of VCAM-1 induced by TNF-α. (**A**) HBMVECs were pretreated with various doses of oleracones 1 h prior to treatment with TNF-α for 8 h and then further incubated with HL-60 cells for 1 h. After non-adherent cells were removed, adherent cells were counted. Scale bar, 100 μm. (**B**,**C**) HBMVECs were pretreated with various doses of oleracones 1 h prior to treatment with TNF-α for 8 h. The ICAM-1 (**B**) and VCAM-1 (**C**) protein levels were determined by Western blot analysis using 20 μg of cell lysates. Quantification was performed using densitometry (ImageJ software, 1.49 v). Results were normalized to β-actin. Data presented are mean ± SEM (n = 3). Statistical difference was determined by Student‘s *t*-test, and the data are presented as the mean ± SEM, * *p* < 0.05, ** *p* < 0.01, *** *p* < 0.001.

**Figure 3 ijms-24-02056-f003:**
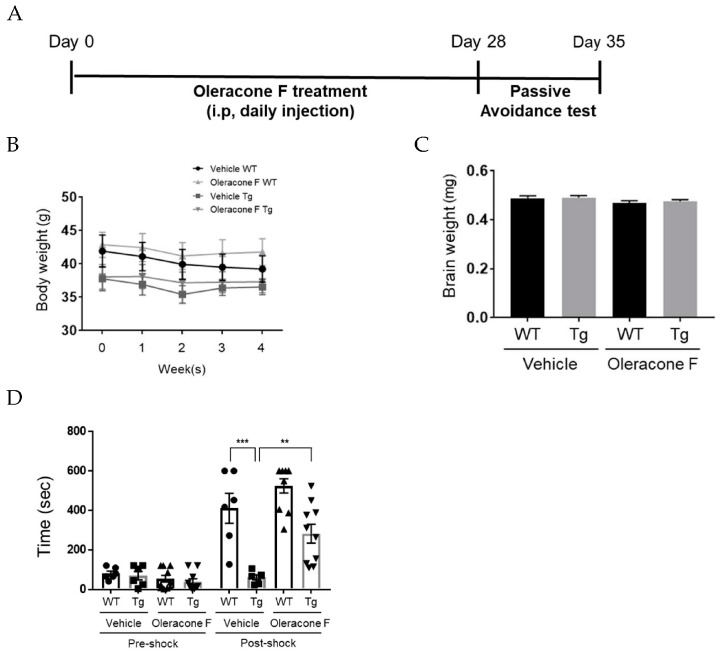
The administration of oleracone F attenuated cognitive impairment in AD transgenic mice. (**A**) Experimental designs are depicted. Wild-type (WT) and AD Tg mice were injected with vehicle (saline) or oleracone F (5 μg/kg, i.p) for 4 weeks. (**B**) The body weight of WT mice was slightly heavier than Tg mice, but no difference between oleracone F-treated and untreated was found. (**C**) Total wet brain weight was measured. (**D**) Passive avoidance test was performed as described in Materials and Methods. The data are presented as the mean ± SEM. Statistical difference was determined by Student’s *t*-test, and the data are presented as the mean ± SEM (n = 6–10), ** *p* < 0.01, *** *p* < 0.001.

**Figure 4 ijms-24-02056-f004:**
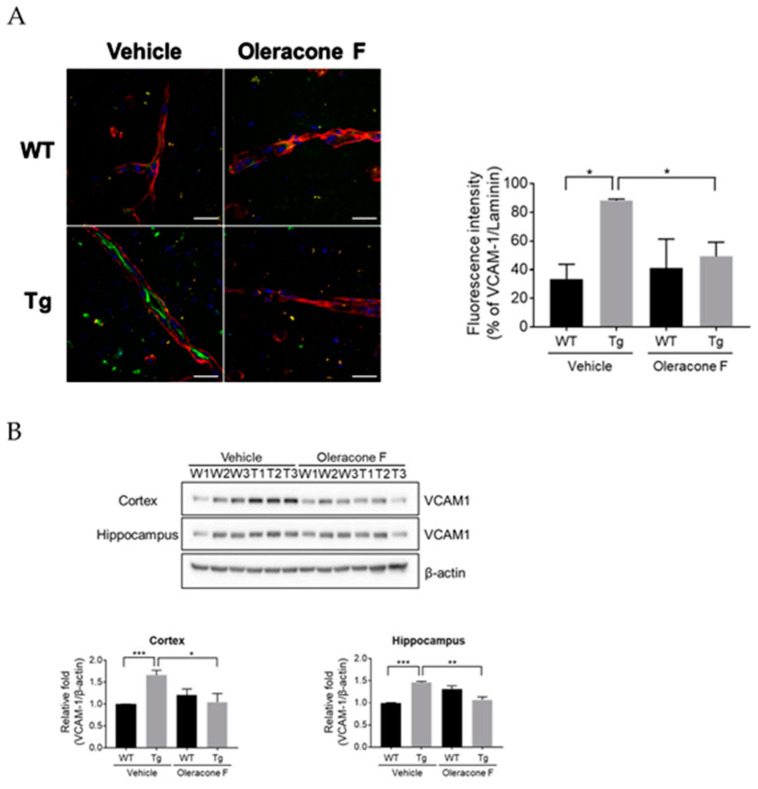
The expression of VCAM-1 is induced in AD transgenic mice, but this increase was blocked by oleracone F. WT and Tg mice were injected with saline or oleracone F (5 μg/kg, i.p) for 4 weeks. (**A**) Sections from mouse brains were immunostained with antibodies against VCAM-1 (red) and laminin (green). Scale bar, 50 μm. Quantifications of the VCAM-1 and laminin immunoreactivities were performed using densitometry (ImageJ software, 1.49 v). Bar graphs of VCAM-1 or laminin represent mean ± SEM (n = 4). (**B**) The VCAM-1 protein levels were determined by Western blot analysis using 30 μg of protein extracts obtained from the mouse cortex and hippocampus. Quantifications of the bands were performed using densitometry (ImageJ software, 1.49 v). Results were normalized to β-actin. Data represented are mean ± SEM (n = 3). Statistical significances are denoted as * *p* < 0.05, ** *p* < 0.001, and *** *p* < 0.001.

**Figure 5 ijms-24-02056-f005:**
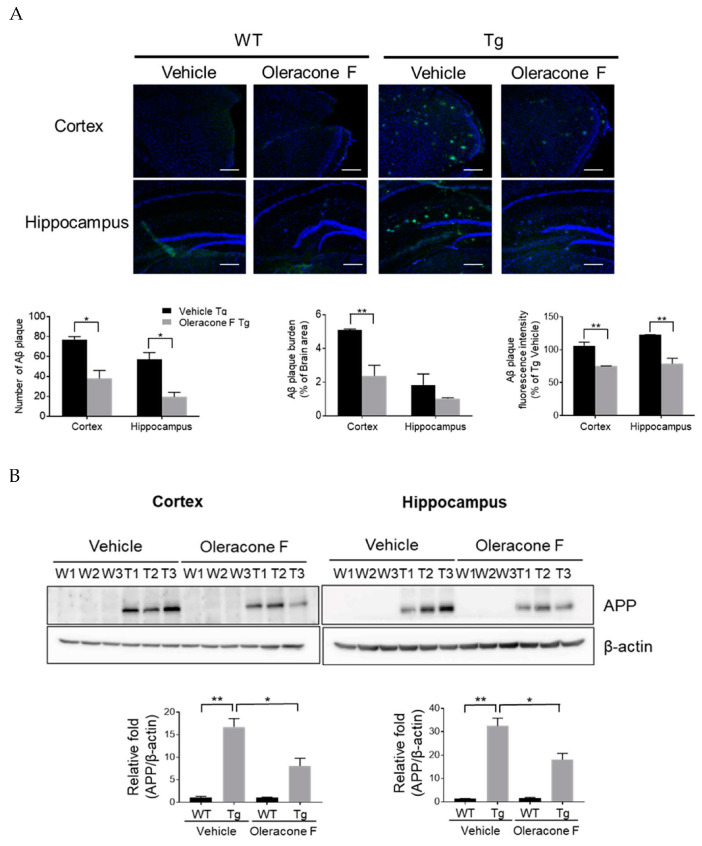
The treatment of AD transgenic mice with oleracone F reduced the brain Aβ levels. WT and Tg mice were injected with saline or oleracone F (5 μg/kg, i.p) for 4 weeks. (**A**) To visualize the amyloid plaques, sections from mouse brains were stained with anti-Aβ antibody (4G8) by the immunofluorescence staining. The number of Aβ plaques, the area occupied by Aβ plaques (Aβ plaque burden), and the fluorescence intensity of Aβ plaques were analyzed with ZEN2009 software on a Zeiss LSM confocal microscope (Carl Zeiss, Jena, Germany). Scale bar, 200 μm. (**B**) The APP protein levels were determined by Western blot analysis using 30 μg of protein extracts obtained from the mouse cortex and hippocampus. Quatifications of the bands were performed using densitometry (ImageJ software, 1.49 v). Results were normaliozed to β-actin. Data represented are mean ± SEM (n = 3). Statistical significance are denoted ad * *p* < 0.05 and ** *p* < 0.01.

**Figure 6 ijms-24-02056-f006:**
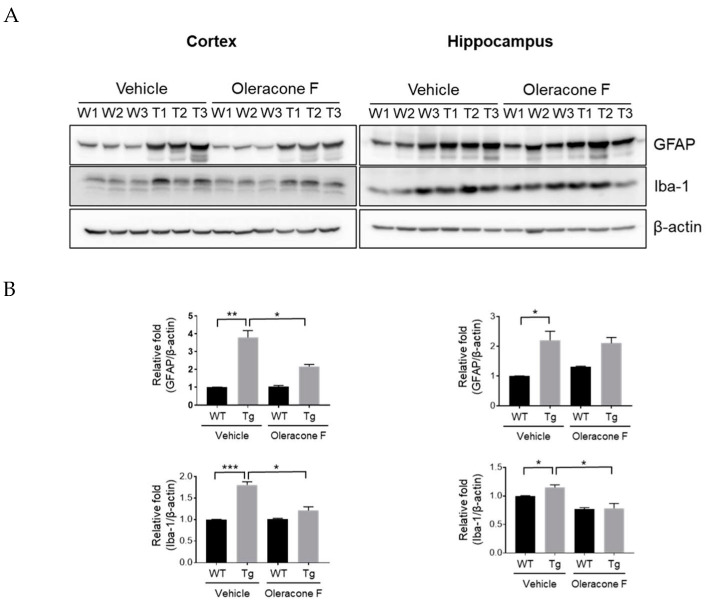
(**A**) The treatment of AD transgenic mice with oleracone F reduced brain inflammation. WT and Tg mice were injected with saline or oleracone F (5 μg/kg, i.p) for 4 weeks. The GFAP and Iba-1 protein levels were determined by Western blot analysis using 30 μg of protein extracts obtained from the mouse cortex and hippocampus. (**B**) Quantifications of the bands were performed using densitometry (ImageJ software, 1.49 v). Results were normalized to β-actin. Data represented are ±SEM (n = 3). Statistical significance are denoted ad * *p* < 0.05, ** *p* < 0.01, and *** *p* < 0.001.

**Table 1 ijms-24-02056-t001:** List of antibodies for Western blot analysis and immunofluorescence staining.

Antibodies	Host	Company	Catalog Number
Anti-ICAM-1	Mouse monoclonal	Santa Cruz	sc8439
Anti-VCAM-1	Rabbit polyclonal	Abcam	ab134047
Anti-Laminin	Rabbit polyclonal	Abcam	ab11575
Anti-Iba1	Rabbit polyclonal	Wako	019-19741
Anti-GFAP	Rabbit polyclonal	Abcam	ab53554
Anti-GFAP	Rabbit polyclonal	Dako	Z0334
Anti-β-amyloid (6E10)	Mouse monoclonal	Biolegend	SIG-39320
Anti-β-amyloid (4G8)	Mouse monoclonal	Biolegend	SIG-39200
Anti-BACE1	Rabbit polyclonal	Abcam	Ab2077
Anti-β-actin	Mouse monoclonal	Abcam	Ab6276

## Data Availability

Not applicable.

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
