# Peer review of "Oleracone F Alleviates Cognitive Impairment and Neuropathology in APPswe/PSEN1dE9 Mice by Reducing the Expression of Vascular Cell Adhesion Molecule and Leukocyte Adhesion to Brain Vascular Endothelial Cells"

_ijms, 2023, doi:10.3390/ijms24032056_

Round 1
Reviewer 1 Report
The paper explores its anti-AD activity and its possible mechanism of action through the detection of different indicators. This study is of reference value.
However, the structure of the paper seems to be incomplete and there is no conclusion section, which is an essential part for a research paper and is recommended to be added.
In addition, the language of the thesis needs further revision and improvement, for example, the definite article is missing in the first sentence of the abstract.
Reviewer 2 Report
Overall, this paper is written in professional English with sufficient introduction and solid data. The idea of this review is worth to be considered to publication after the authors make some necessary changes in order to deliver a more approachable and more friendly-reading manuscript.
Line 405-410. In which concentration was used DMSO, having in view its known toxic potential on cell viability. Explain also why intraperitoneal route of administration was chosen.
Line 83-93. Having in view that the authors worked with 3 oleracones, to better characterise and compare the cellular toxicity of oleracones the IC50 value calculation is very important. So, I encourage the authors to calculate this parameter for a better discussion regarding the influence of compounds on human microvascular endothelial cell viability.
The authours should add a brief paragraph for conclusions of their study, in according to the obtained results.
Reviewer 3 Report
A very good paper. Congratulations to the authors.
A brief, one phrase characterization should be made of the Tg (AD model mice). What model? How do you evaluate cognitive decline and over what periods of time? These data are necessary, as :
-
Not all readers are familiarized with neurodegenerative transgenic models
-
what is the spontaneous evolution of Tg compared with wild -type and over what time?
-
A clearer comparison between the behavioral evolution of non-treated Tg mice with wild-type would be welcome
I do not know if inserting the Methods chapter after the discussions is a new trend nowadays (I have seen it several times in papers ) but I consider it a very unhappy move. Discussing results before sharing how the work was done is a fine way of disorienting the reader and prompting questions that are answered later, but in an unlikely setting, thus disturbing the understanding of the paper.
Also, a very brief heading of conclusions is very useful at the end of the paper, as it will increase the impact of the results. Ending dryly with the statistics is a good way for compelling the reader to return to the middle, for reading the discussions again. When researchers have to read several papers a day, not a good move for citations…
